# Hydrological Impact of the New ECMWF Multi-Layer Snow Scheme

**Ervin Zsoter** [1,2,*], **Gabriele Arduini** [1], **Christel Prudhomme** [1], **Elisabeth Stephens** [3,4] and **Hannah Cloke** [2,3,5,6]

1    European Centre for Medium-Range Weather Forecasts, Shinfield Park, Reading RG2 9AX, UK; gabriele.arduini@ecmwf.int (G.A.); christel.prudhomme@ecmwf.int (C.P.)
2    Department of Geography and Environmental Science, University of Reading, Reading RG6 6AB, UK; h.l.cloke@reading.ac.uk
3    Department of Meteorology, University of Reading, Reading RG6 6BB, UK; elisabeth.stephens@reading.ac.uk
4    Red Cross Red Crescent Climate Centre, 2593 HT The Hague, The Netherlands
5    Department of Earth Sciences, Uppsala University, 752 36 Uppsala, Sweden
6    Centre of Natural Hazards and Disaster Science (CNDS), 752 36 Uppsala, Sweden
*    Correspondence: ervin.zsoter@ecmwf.int

**Abstract:** The representation of snow is a crucial aspect of land-surface modelling, as it has a strong influence on energy and water balances. Snow schemes with multiple layers have been shown to better describe the snowpack evolution and bring improvements to soil freezing and some hydrological processes. In this paper, the wider hydrological impact of the multi-layer snow scheme, implemented in the ECLand model, was analyzed globally on hundreds of catchments. ERA5-forced reanalysis simulations of ECLand were coupled to CaMa-Flood, as the hydrodynamic model to produce river discharge. Different sensitivity experiments were conducted to evaluate the impact of the ECLand snow and soil freezing scheme changes on the terrestrial hydrological processes, with particular focus on permafrost. It was found that the default multi-layer snow scheme can generally improve the river discharge simulation, with the exception of permafrost catchments, where snowmelt-driven floods are largely underestimated, due to the lack of surface runoff. It was also found that appropriate changes in the snow vertical discretization, destructive metamorphism, snow-soil thermal conductivity and soil freeze temperature could lead to large river discharge improvements in permafrost by adjusting the evolution of soil temperature, infiltration and the partitioning between surface and subsurface runoff.

**Keywords:** land-surface modelling; snow scheme; hydrological processes; river discharge; surface runoff; permafrost

## 1. Introduction

Land-surface models (LSMs) are vital tools for simulating water and energy fluxes at the land–atmosphere interface of the Earth [1]. Although LSMs were originally designed to provide lower-boundary conditions to the atmosphere [2], with the improving realism of these models they are increasingly used for simulating the hydrological cycle [3] and supporting hydrological applications (e.g., [4–6]).

However, there are still significant limitations in the representation of the hydrological cycle in LSMs, as important processes can still be inadequately modelled or even neglected for runoff generation, for instance groundwater simulation, snow-vegetation interactions, representation of frozen soil and lateral flow between adjacent grid cells among others [1,5,7–11].

Simulating the extent and variability of the snow cover is a crucial aspect of land-surface modelling, as it strongly influences the energy and water balances [12,13]. Snow schemes have various complexities in the representation of the snow physics [14–18],

differing largely in their handling of the snowpack and the creation of snowmelt, which in turn impacts runoff generation and river flow in snow dominated areas [8,19].

The snow scheme currently used operationally in the ECLand land-surface model at the European Centre for Medium-Range Weather Forecasts (ECMWF) [20,21] and which is used in the production of the ERA5 [22] and ERA5-Land [23] reanalysis datasets, is a single-layer snow scheme (SLS hereafter) with an additional snow layer on top of the soil [17]. The use of only one layer limits the handling of the temporal evolution of the snow, as changes on multiple time scales (i.e., diurnal to seasonal) cannot be accurately represented. This has a significant impact on the quality of the snow depth, mainly during periods of accumulation and ablation, which then impacts the soil freezing, the snowmelt and ultimately the hydrological cycle [8,24].

Snow schemes using multiple layers to represent the snowpack offer significant improvement on the single-layer schemes with better handling of the snow processes. For hydrological and climate applications, so-called "intermediate complexity" snow schemes are generally used, following the terminology introduced by Boone and Etchevers [14]. Such schemes include a description of snowpack properties, such as density and temperature, using a limited number of vertical layers, as opposed to detailed snow physics models, which aim at simulating the microstructure properties of the snowpack as well (see for instance Vionnet et al. [25]). Such intermediate complexity schemes have been implemented in various LSMs used in Earth system modelling during the last decade. Examples are ECLand [8,26], Noah [24], JULES [27], ISBA [14,16] and the ORCHIDEE [18] land-surface models.

The multi-layer snow scheme (MLS hereafter), introduced experimentally in ECLand, is an intermediate complexity scheme representing the vertical structure and evolution of snow temperature, density, liquid water content and surface snow albedo with a maximum of five layers [26]. It has been shown to increase the realism of the snow representation, including decreasing snow depth and snowmelt timing errors, and has been shown to largely improve 2-metre temperature in coupled forecasts, especially in clear sky conditions [26].

The more realistic snowpack representation, with better snow water equivalent, snow depth and snowmelt, is expected to improve the hydrological cycle and thus have a positive impact on river flow simulations [28]. Preliminary studies have demonstrated improvements in localized settings [8,18] or combined with other land-surface improvements [29]. However, the hydrological impact analysis of MLSs, focusing on river flow, has not been done at regional or global scales.

Areas in which snow plays an important role are predominantly found in the Northern Hemisphere over higher latitudes. A large fraction of these areas is permafrost [30], where soil freezing/thawing conditions play a major role in controlling the hydrological processes [31,32]. Representing permafrost in LSMs is important for better understanding of the hydrological variability and the impacts of climate change [32–34].

In this study, the hydrological impact of the MLS implemented in ECLand is analyzed on more than 400 catchments globally, with over a third located partially or entirely in permafrost areas. To achieve this, ECLand experiments forced with ERA5 over the period 1979–2018 are coupled to the Catchment-based Macro-scale Floodplain model (CaMa-Flood; [35]) to generate river discharge, allowing direct comparison with gauged observations. Different sensitivity experiments are conducted to evaluate the impact of the more physically complex snow scheme on the terrestrial hydrological processes, with particular focus on permafrost, where complicated error dynamics arise from different land-surface processes. Two main questions are posed:

- How does the MLS impact the simulated hydrological processes and river discharge, especially in the snowmelt-driven flood season?
- How sensitive is the hydrological representation of permafrost to the snow and soil parametrization?

## 2. Materials and Methods

In this section, the data set, models and methods used will be described.

### 2.1. ECLand Land-Surface Model and Offline Methodology

The hydrological core of the analyzed data sets was provided by the ECLand land-surface model, formerly known as HTESSEL (The Hydrology-Tiled ECMWF Scheme for Surface Exchange over Land [20,21,36]). ECLand is part of the Integrated Forecasting System (IFS) at ECMWF and used in coupled land-atmosphere simulations for describing the evolution of soil, vegetation and snow conditions over land, at various spatial resolutions, from short- to seasonal-range.

In ECLand, up to six tiles are present over land (bare ground, low and high vegetation, intercepted water, shaded and exposed snow) and three over water (inland, open and frozen water) that provide the interface between the atmosphere and the one-dimensional soil column, with all tiles having their separate energy and water balances.

The snowfall (the solid fraction of precipitation) is collected in the snowpack, which overlays the soil [20]. The fraction of the soil that is covered by snow (snow cover fraction) is parametrized as a linear function of snow depth [17]. This assumes that a model grid-box is fully covered for snow depth greater than 10 cm. The same parametrization is used for exposed and shaded snow (i.e., snow under high vegetation) tiles in ECLand. The soil is divided into four layers with fixed layer depths (0–7, 7–28, 28–100 and 100–289 cm). Runoff is generated as fast (surface) and slow (subsurface) components at each grid point [20,21].

Snowmelt occurs when the temperature of the snow is high enough, contributing to surface runoff, soil infiltration and evaporation. Some part of rain and snowmelt will be removed as surface runoff. This surface runoff fraction depends on the standard deviation of the sub-grid scale orography (a measure of unresolved orographic features), the soil texture and the soil water content. Subsurface runoff is the water leaving the soil column at the bottom. It depends on the infiltration and surface evaporation as top boundary conditions, while water can be extracted by roots in each soil layer where vegetation is present [20,21].

ECLand can be used in a stand-alone mode, when the model runs uncoupled from the atmosphere, usually with hourly time step, forced with near-surface meteorological input data of temperature, specific humidity, wind speed, surface pressure, radiative fluxes (downward solar and thermal radiation) and water fluxes (liquid and solid precipitation), without land data assimilation. This offline research methodology provides an affordable way of testing land-surface improvements and has been used in various applications (e.g., [26,37–39]). The ERA5-Land dataset is a prime example of this methodology, which was produced as an offline ECLand simulation with downscaled ERA5 meteorological forcing on higher resolution, including an elevation correction for the thermodynamic near-surface state [23].

### 2.2. ERA5 Reanalysis

The meteorological forcing for the offline ECLand simulations was taken from ERA5, the latest global climate reanalysis of ECMWF [22]. ERA5 is a key contribution to the EU-funded Copernicus Climate Change Service (C3S) and is open access and free to download for all users (https://cds.climate.copernicus.eu/ (accessed on 24 October 2021)). It covers the period 1979 to present, with a preliminary version also available from 1950. It includes a high-resolution component (~31 km) and a lower resolution (~62 km) ensemble component with 10 members. In this study, the high-resolution component (hereafter referred to as ERA5) was used from 1979 with ~31 km horizontal resolution and hourly output frequency.

### 2.3. CaMa-Flood River-Routing

The hydrodynamics to produce river discharge from the ECLand runoff output were simulated by CaMa-Flood [35], a global river-routing model, which is part of ECLand since IFS cycle 47r1 [21]. CaMa-Flood routes runoff generated by land-surface models to

oceans or inland seas. The model calculates river and floodplain water storages, discharge, water depth, as well as flood inundation. CaMa-Flood does not currently include the representation of dams and permanent lakes and wetlands are only treated as part of the floodplain storages. CaMa-Flood is computationally cheap to run and has been used widely in global climatological research studies, such as Emerton et al. [40], Dottori et al. [41] and Zsoter et al. [39].

### 2.4. ECLand Snow and Soil Freezing Schemes

The current SLS, used operationally in ECLand, is a basic energy balance model describing the temporal evolution of the heat and mass contents of the snowpack [17]. The MLS, used experimentally in ECLand, is an intermediate complexity snow scheme [14], which represents the vertical structure and time evolution of snow temperature, density, liquid water content and surface snow albedo with up to five active snow layers. In this section, the important model features are described that are necessary to understand the scheme variations tested in this paper (described later in Section 2.8). Further details of MLS, including a detailed comparison of snowpack properties to SLS, are given in Arduini et al. [26].

#### 2.4.1. Snow Vertical Discretization

The number of active snow layers and their associated thicknesses are defined diagnostically at each time step before the updating of the other snow variables. The number of active layers ($N$) is dependent on the snow depth $D_{sn}$ and varies from one layer to a maximum of five ($N_{max}$). The topmost snow layer, in contact with the atmosphere, is assumed to be the first one, whereas layer N is the one in contact with the soil. $N$ is determined as the lowest number that satisfies the following inequality for $N = 1, \ldots, N_{max}$-1:

$$\sum_{j=1}^{N+1} D_{min,j} > D_{sn}, \tag{1}$$

where $D_{min,j}$ is the minimum snow depth allowed for layer $j$. It is by default set to 0.05 m for all layers (denoted hereafter as $D_{min}$ for all layers). The depth of the first layer is defined as:

$$D_{sn,1} = \begin{cases} D_{sn}, & if\ D_{sn} < 2D_{min} \\ D_{min}, & if\ D_{sn} > 2D_{min} \end{cases}. \tag{2}$$

Note that with this choice, MLS has only one active snow layer for $D_{sn} < 0.1$ m. For the remaining layers, the vertical discretization is defined as:

$$D_{sn,i=2,...,N_{max}} = \begin{cases} 0, & if\ D_{sn} < \sum_{j=1}^{j=i} D_{min,j} \\ min\left[\frac{D_{sn}-D_{sn,1}}{N-1}, D_{max,i}\right], & if\ D_{sn} > \sum_{j=1}^{j=i} D_{min,j} \end{cases}, \tag{3}$$

where $D_{max,i}$ is the maximum snow depth allowed for the $i$th active snow layer. This effectively means, the snow is evenly divided into the remaining layers, as long as the maximum layer depths allow it. By default, these maximum values are 0.05, 0.10, 0.20, $\infty$ and 0.15 for layers $1-N_{max}$, respectively. This definition of the maximum layer depths means that when all $N_{max}$ snow layers are active, the $N_{max} - 1$ layer is used as the accumulation layer for thick snowpacks. This layering allows a relatively high vertical resolution both at the interfaces to the atmosphere above and to the soil underneath the snowpack. Take $D_{sn} = 1.25$ m as an example, for this depth the snowpack is discretized into 5 layers with thicknesses of 0.05, 0.1, 0.20, 0.75 and 0.15 m from top to bottom.

#### 2.4.2. Destructive Metamorphism of the Snow

The density of freshly fallen snow can vary rapidly with time due to metamorphic processes, that is, the change in shape and size of snow grains once they settle in the snowpack. The rate of change of snow density due to destructive metamorphic processes

of the snow ($\xi$) is parametrized in both SLS and MLS using the formulation introduced by Anderson [42]:

$$\frac{1}{\rho_{sn,i}} \left[ \frac{\partial \rho_{sn,i}}{\partial t} \right]_{\xi} = a_{\xi} \exp\left[ -b_{\xi}\left( T_f - T_{sn,i} \right) - c_{\xi} \max\left( 0, \, \rho_{sn,i} - \rho_{\xi} \right) \right], \tag{4}$$

where $T_{sn,i}$ and $\rho_{sn,i}$ are the snow temperature and snow density for each snow layer (for SLS, $i = 1$) and $a_{\xi} = 2.8 \times 10^{-6}$ s$^{-1}$, $b_{\xi} = 4.2 \times 10^{-2}$ K$^{-1}$, $c_{\xi} = 460$ m$^3$kg$^{-1}$, $T_f = 273.16$ K and $\rho_{\xi} = 150$ kg m$^{-3}$.

### 2.4.3. Snow-Soil Thermal Conductivity

In SLS and MLS, the thermal coupling between the snowpack and the soil underneath is described using a thermal conductance between the two media ($\lambda_b$), thus the heat flux ($G_b$) can be written as:

$$G_b = \lambda_b \left( T_{sn,N}, \, - T_{so} \right), \tag{5}$$

where $T_{sn,N}$ is the snow temperature of the bottom active snow layer ($N = 1$ for SLS) and $T_{so}$ is the temperature of the topmost soil layer. Given that the heat resistances are in series, the sum of the inverse of the conductance of each medium yields the total conductance of the snow-soil system, that is:

$$\lambda_b^{-1} = \frac{l_b \Delta z_{sn,N}}{\lambda_{sn,N}} + \frac{l_b \Delta z_{so}}{\lambda_{so}}, \tag{6}$$

where $\Delta z_{sn,N}$ and $\lambda_{sn,N}$ are the thickness and conductivity, respectively, of the bottom active snow layer, and $\Delta z_{so}$ and $\lambda_{so}$ are the thickness and conductivity of the topmost soil layer. The parameter $l_b$ is set as 0.5 in SLS, whereas it was changed to 1 in MLS, as described in Arduini et al. [26], to account for the additional insulation effect due to organic material or vegetation between the snow layer and the soil.

### 2.4.4. Soil Freezing Scheme and Relationship to Runoff Generation

Frozen soil is characterized by very different thermal and hydrological properties compared to unfrozen (wet) soil. For instance, precipitation infiltrates less into a frozen soil, thus more water goes into runoff than into the soil. In ECLand, a simplified representation of unfrozen soil was introduced by Viterbo et al. [43] to account for the latent heat release/absorption of soil water around 0 °C, reducing a pronounced cold bias in 2-metre temperature in the ECMWF forecasts. In this simplified approach, the frozen water fraction in each soil layer ($\theta_{ice,i}$, $i = 1, \ldots, 4$) is given as a function of the soil temperature ($T_{so,i}$) as:

$$\theta_{ice,i} = \begin{cases} 0 & for \ T_{so,i} > T_{th} \\ 0.5\left( 1 - sin\left( \frac{\pi\left( T_{so,i} - 0.5\left( T_{Th} + T_{Fr} \right) \right)}{T_{Th} - T_{Fr}} \right) \right) & for \ T_{Fr} \leq T_{so,i} \leq T_{Th}, \\ 1 & for \ T_{so,i} < T_{Fr} \end{cases} \tag{7}$$

where $T_{Th}$ = +1 °C (thaw temperature) and $T_{Fr} = -3$ °C (freeze temperature) are the soil temperature thresholds for which soil water is totally unfrozen or frozen, respectively, and $[T_{Th}, T_{Fr}]$ is the temperature interval for which phase change can occur. Both SLS and MLS use this soil freezing scheme.

The runoff generation in EC-Land follows the formulation described in Balsamo et al. [20], which includes a dependency of surface runoff on soil textures as well as sub-grid scale orography features not resolved at the resolution used in the simulation. The surface runoff can be as large as 50% of the available precipitation and snowmelt, for large standard deviation of the sub-grid scale orography and finer soil textures.

When the soil is partially frozen, the surface runoff is enhanced, as less water infiltrates and percolates within the soil column. This soil freezing mechanism is represented by

reducing the soil hydraulic conductivity and diffusivity. This is done by computing the soil hydraulic conductivity and diffusivity as a weighted average of the values for the unfrozen soil water fraction and for the frozen water fraction [20].

As previously stated, the frozen water fraction parametrization described in Equation (7) has been developed to address temperature errors in weather forecasting applications. Such simplified approach can have limitations for hydrological applications in cold regions, where the interaction between frozen soil and runoff is key in modulating river streamflow (see for instance [44]).

*2.5. River Catchment Selection*

The river catchments were selected for this study only if they experience regular snowfall and they have adequate river discharge observations available. Observations are selected from the Global Runoff Data Centre (GRDC; https://www.bafg.de/GRDC/ (accessed on 20 November 2019)) supplemented by additional data collected by the Copernicus Emergency Management Service for floods in the Global flood Awareness System (GloFAS), as described in Harrigan et al. [6]. The catchments and associated river discharge observations are selected with the following set of criteria:

- Minimum 8 years of river discharge observations in 1980–2018. Gaps are not considered a problem, as long as the climatological mean can be computed for each day of the year (see Section 2.7);
- Stations in snow impacted climate, defined by the percentage ratio of ERA5 snowfall and total precipitation being at least 10%, based on the 1979–2018 mean for each catchment;
- Catchment area of at least 5000 km$^2$ (e.g., minimum of 8 river pixels);
- Good general quality. After visual inspection of the river discharge time series, the catchments that showed observation errors, problems with station metadata (wrong or uncertain location, etc.) or visible influence of dams and lakes were excluded. To help with identifying reservoir and lake influence, the Global Reservoir and Dam Database (GRAND; [45]) and the Global Lakes and Wetlands Database (GLWD; [46]) were used as visual tools.

Out of the 2119 stations in the Copernicus Emergency Management Service GloFAS station database with at least 1 year of observations, 1913 met the criteria of at least 8 years of data, 889 catchments had also at least 10% snowfall ratio and 849 were additionally over 5000 km$^2$ area. In total, 453 catchments were selected after the quality checks, almost entirely in the Northern Hemisphere (Figure 1). For the sensitivity analysis in the permafrost, a specific area was defined in the 60–80 N belt containing parts of northern Siberia, Alaska and western Canada. It focuses on the coldest parts of the permafrost. The eastern area of Canada was omitted due to the large number of lakes in the area to avoid unduly influencing of the overall results. Figure 1 shows the bounding box, together with the permafrost, as defined by areas of below 0 °C climatological annual mean temperature in the lowest soil layer of ECLand in ERA5 for 1st of June. Within ECLand, the land use of the catchments in the permafrost sensitivity area are mainly a mixture of tundra and boreal forest vegetation, whereas the soil textures are medium-coarse and coarse soil types, which are characterized by a relatively low field capacity and higher hydraulic conductivity (for details on land use and soil hydraulic properties see [20,21]).

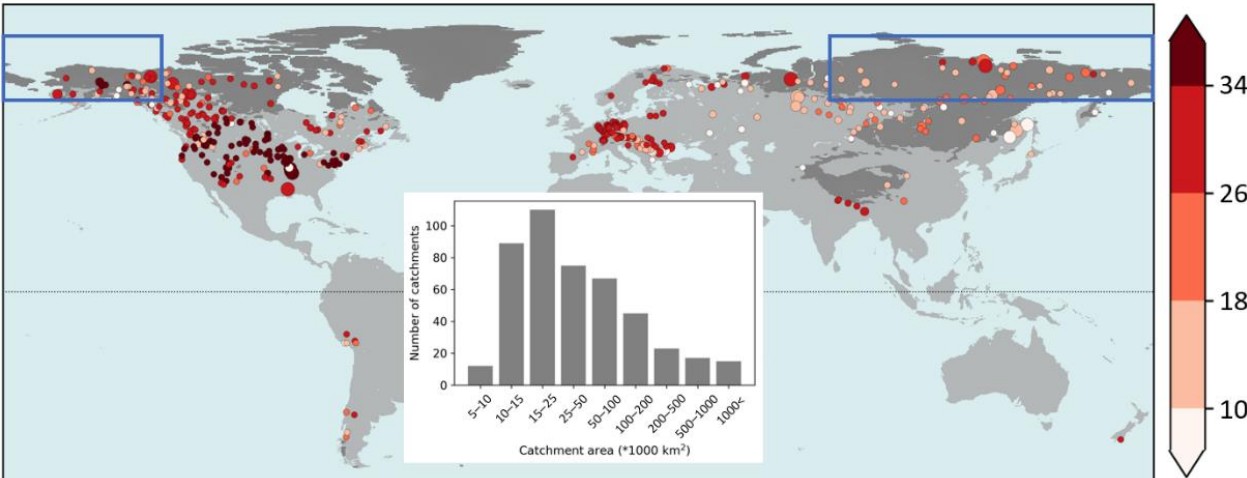

**Figure 1.** Stations used in this study with the number of river discharge observation years available (8 years is the minimum). In total, 453 catchments worldwide. The darker grey shading indicates the permafrost areas (defined as the area where the lowest soil layer's temperature is below 0 °C in the ERA5 climate mean on 1st of June, based on 1980–2018), while the blue rectangle shows the sensitivity area defined in the permafrost. The distribution of the catchment upstream area values is provided in the inset table (please note the area values are divided by 1000).

### 2.6. Verification Statistics

Hydrological performance is assessed with the modified Kling–Gupta efficiency (*kge*; [47,48]). The *kge* is increasingly considered as the standard performance metric in hydrology [49,50]. It can be decomposed into three components, measuring the correlation, bias and variability errors, which makes it an easy to interpret metric, ideal for assessing hydrological dynamics:

$$kge = 1 - \sqrt{(pcorr - 1)^2 + \left(\frac{\mu_s}{\mu_o} - 1\right)^2 + \left(\frac{\sigma_s/\mu_s}{\sigma_o/\mu_o} - 1\right)^2}. \tag{8}$$

In the *kge* decomposition, *pcorr* is the Pearson correlation coefficient between daily simulation (*s*) and observation (*o*) time series, measuring the temporal errors; $\mu$ is the mean and $\sigma$ is the standard deviation of the time series. In Equation (1), $\frac{\mu_s}{\mu_o}$ is the bias ratio, while $\frac{\sigma_s/\mu_s}{\sigma_o/\mu_o}$ is the variability ratio, which highlight how close the mean and the variability (normalized by the means) are in the simulated and observed time series. The *kge* and its three components are all dimensionless. The correlation ranges from −1 to +1, with +1 showing perfectly strong linear relationship and 0 no linear relationship (−1 being perfect inverse relationship). In the *kge* definition, the bias and variability ratios both range from 0 to infinity, with 1 being the optimal value. The bias and variability errors, used in this study, were defined by the bias and variability ratio components of the *kge* as follows:

$$bias = \frac{\mu_s}{\mu_o} - 1 \text{ and } var = \frac{\sigma_s/\mu_s}{\sigma_o/\mu_o} - 1, \tag{9}$$

This way, the optimal score value transforms to 0, highlighting the direction of the biases more intuitively, with negative values showing underprediction, while positive ones overprediction. To aid comparison between different experiments, the absolute version of the *bias* (*abias*) and *var* (*avar*) errors are also used. Change in these metrics can directly highlight improvement or deterioration, while they also share the optimal value of zero.

### 2.7. Daily Climatology Computation

The land-surface contribution to the water budget is diagnosed qualitatively at specific catchments by using daily climatological mean time series of simulated and observed river discharge, runoff (surface and subsurface components), snowpack water content and snowmelt, evaporation, soil temperature and water content (at different layers) and precipitation, computed from daily values in the 1980–2018 period. All water related variables are converted into catchment totals as the sum of all the grid point values in the catchment, in order to compare them directly to river discharge. For temperature variables, the catchment averages are used.

These climatological means are computed for every day of the year (1 January–31 December), by applying a 21-day window, centered over the day of the year. To aid direct comparison with observed river discharge, only those days of 1980–2018 are considered for the climatological mean computation, which have river discharge observations available. For the climatological mean, the minimum data length was lowered to 4 years, in order to maximize the likelihood of being able to compute the mean for every day of the year. With this choice, the climate sample size could range from 84 values (4 years, 21 values each) to over 800 (most years in 1980–2018) to compute the daily climate mean.

### 2.8. Experimental Setup

In this study, the hydrological impact of the MLS is analyzed on ECLand/CaMa-Flood coupled experiments. The runoff is produced by ECLand, while the CaMa-Flood model is used to produce river discharge by routing the runoff over the 15 arcmin (~25 km on the Equator) river network, which is an appropriate horizontal resolution for the related meteorological forcing of ERA5.

CaMa-Flood uses a 1-h time step and a 24-h output frequency to match the 24-h reporting frequency of the river discharge observations. All experiments are generated for the ERA5 period of 1979–2018, while for the hydrological analysis 1980–2018 is used with 1979 omitted to account for the spin-up in the simulations. Preliminary analysis showed that a 1-year spin up period is appropriate, as a longer spin up period did not have a large influence on results but considerably reduced the sample size.

In total, 13 experiments are produced and compared (Figure 2). Two experiments use SLS, while the other 11 are produced with variations of MLS and the soil freezing parametrization in ECLand, for analyzing the hydrological sensitivity, focusing on permafrost areas. The very high computational cost of running these experiments meant that it was not possible to run all permutations of the schemes and parameters tested, instead possible modifications build on each other incrementally.

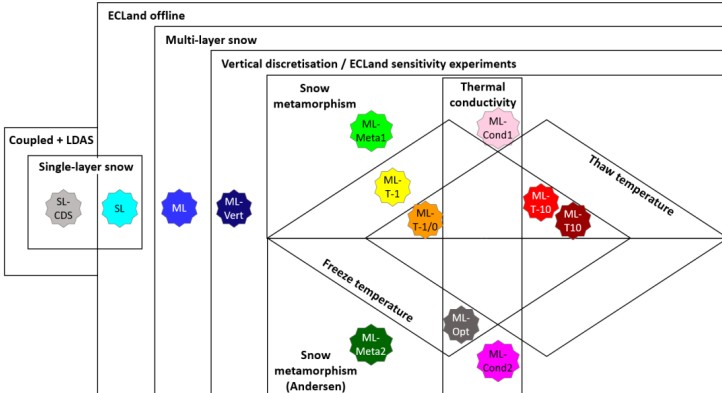

**Figure 2.** Experiments analyzed in this study, describing the main simulation features including the snow and soil scheme modifications for sensitivity analysis in permafrost. The experiments are displayed with the star shapes, using a short name and a color to help identifying them throughout the paper.

**Single-layer, online, fully coupled with land data assimilation: SL-CDS**

The first experiment involves a single CaMa-Flood run, using the original ERA5 runoff data (downloaded from the Copernicus Climate Data Store), which is produced online with land-atmosphere coupling, including atmospheric and land data assimilation and SLS (SL-CDS).

**Offline experiments**

All other experiments that follow include a surface only (offline) ECLand simulation, without land-atmosphere coupling and land data assimilation, to produce runoff, which is then routed with CaMa-Flood. This is because online experiments with coupling and data assimilation would be infeasible to run due to the very high computational cost. The offline experiments are initialized from the ERA5 state on 1 January 1979 and forced with ERA5 near-surface meteorological data on ~31 km horizontal resolution and hourly output frequency (see Section 2.1 for further details).

**Single-layer snow scheme: SL**

In order to compare the online and offline modelling approaches directly, one of the offline experiments is run with the SLS (SL), to be compared with SL-CDS (see Zsoter et al. [39] for further details on online/offline comparison).

**Multi-layer snow scheme: ML**

The first MLS experiment uses the default snow parametrization, introduced by Arduini et al. [26] and default ECLand soil freezing parametrizations (see Section 2.4) and can be considered the default multi-layer experiment (ML).

**ECLand sensitivity experiments**

The following 10 experiments are variations of the ECLand snow and soil freezing schemes, designed to evaluate the river discharge sensitivity in permafrost.

**Vertical snow discretization: ML-Vert**

The first change is for the vertical snow discretization in MLS (ML-Vert). It introduces thicker snow layers over complex terrains with deep snowpack and reduces issues related with excessive melting over mountainous regions [21]. Complex terrain is defined as areas with standard deviation of the sub-grid scale orography over 50 m. For flat terrain and for $D_{sn} < 0.25$ m over complex terrain, the same vertical discretisation is used as in ML (described previously in Section 2.4). For $D_{sn} > 0.25$ m over complex terrain, $D_{max,i}$ and $D_{min,i}$ can vary with snow depth in Equations (1)–(3) as follows:

$$D_{min,i} = \min(0.25, \ 0.10 + \alpha_0(D_{sn} - 0.25)) \text{ for } i = 1 \qquad (10)$$

$$D_{max,i} = \begin{cases} \min(0.25, \ 0.10 + \alpha_0(D_{sn} - 0.25)) \text{ for } i = 1 \\ \min(0.30, \ 0.15 + \alpha_0(D_{sn} - 0.25)) \text{ for } i = 2, \ldots, N_{max} \end{cases}, \qquad (11)$$

where $\alpha_0 = 0.1$ is a predefined parameter. For the example of $D_{sn} = 1.25$ m (as in Section 2.4), the thickness of the 5 layers changes to 0.20, 0.25, 0.25, 0.30 and 0.25 m, which means the snowpack is more evenly distributed for this large depth.

**Destructive metamorphism of the snow: ML-Meta1 and ML-Meta2**

The second group of changes is for the snow metamorphism. The parametrization for the destructive metamorphism of the snow (Equation (4)), as used in SL, SL-CDS and ML with the default value of $c_\xi = 460$ m$^3$ kg$^{-1}$, implies that the rate of snow density changes due to the 2nd density-dependent term in the exponential, and is active only for $\rho_{sn,i} < 150$ kg m$^{-3}$, that is for relatively fresh snowpack. Cao et al. [51] pointed out, while evaluating soil temperature and snow characteristics of ERA5-Land in permafrost regions, that this can partly explain the underestimation of the snow density of ERA5-Land. They have argued that the underestimation of the snow density could lead to an overestimation

of the thermal decoupling between the atmosphere and the soil underneath. This could contribute to the warm bias of soil temperature over permafrost regions as less heat is diffused from the soil towards the colder atmosphere above.

To address this, the impact of the representation of the snow density on river discharge is explored in two experiments, by changing the value of the parameter $c_\xi$. In the first experiment (ML-Meta1), the parameter $c_\xi$ is varied for the five snow layers, using values closer to the 0.046 m$^3$ kg$^{-1}$, reported in Anderson [42], as follows:

$$c_\xi = (0.112,\ 0.152,\ 0.192,\ 0.288,\ 0.488)\ \text{m}^3\ \text{kg}^{-1}. \tag{12}$$

These values are chosen as they were the best compromise in terms of land-surface and atmospheric impact (i.e., in particular 2-metre temperature) in coupled land-atmosphere forecast experiments, which were conducted for the foreseen implementation of the MLS in operational weather forecasts at ECMWF. The decreasing values towards the first (top) snow layer imply that the destructive metamorphic process of the snow is more active for the top snow layers, whereas for the settled snow at the bottom of the snowpack it is less active, by this making the compaction process slower.

Another experiment (ML-Meta2) uses the original value of $c_\xi = 0.046$ m$^3$ kg$^{-1}$ for all snow layers, as reported by Anderson [42], which is a commonly used approximation in land-surface models. This latter unified and low value option of $c_\xi$ gives the opportunity to explore the maximum sensitivity to the destructive metamorphic process of the snow in the snow density representation, without considering the implication for the snow-atmosphere coupling (e.g., atmospheric scores), given that the feedback to the atmosphere is not considered in the offline experiments presented in this work. In coupled experiments, the impact of this change would be expected to be substantial on the near surface variables, like 2 m temperature, possibly requiring an additional tuning of other land-atmosphere coupling parameters. Both experiments build on ML-Vert by adding the snow metamorphism corrections to the vertical snow depth discretization adjustment.

**Snow-soil thermal conductivity: ML-Cond1 and ML-Cond2**

The next two experiments use the parameter $l_b = 0.5$ in the snow-soil thermal conductivity computation (Equation (6)), the same value as in SLS and reduced by half compared with MLS. Organic material distribution is highly variable both in horizontal and vertical (within the soil) scales and its handling requires more sophisticated parametrizations [16]. For this reason, in these two experiments we relax the hypothesis of $l_b = 1$ in MLS, removing the additional thermal insulation effects caused by organic material, effectively considering half of the topmost soil layer in the computation of the conductance between the two media (see Equation (6)), consistently with what is done for the other snow-free land-surface tiles of ECLand. Effectively, this increases the thermal coupling between the snow and soil and thus also increases the heat flux between the two media in these experiments. Both experiments include the vertical discretization change (ML-Vert), while ML-Cond1 is run together with the first snow metamorphism change in ML-Meta1 and ML-Cond2 with the change in ML-Meta2.

**Soil freeze and thaw temperatures: ML-T-1, ML-T-1/0, ML-T10 and ML-T-10**

The next step in the experiment design is to test the sensitivity of river discharge simulations to the fraction of frozen water in the soil. Soil freezing and thawing is particularly important during the snowmelt season, when the thermal coupling of the topmost soil layers to the atmosphere increases quickly as the snow melts and phase changes of the soil water can occur. Out of the two temperature parameters, the freeze temperature has a more direct impact on runoff generation during the spring period, whereas the thaw temperature could be more important for energy fluxes and the coupling to the atmosphere in the fall/winter season. Gouttevin et al. [34] suggested that a shorter temperature interval for phase change works better for permafrost.

These soil temperature experiments are run including the changes in ML-Vert and ML-Meta1. First, while keeping the $T_{Th}$ at the default value of +1 °C in Equation (7), the $T_{Fr}$ parameter is increased from the default −3 to −1 °C (ML-T-1). Similarly, in another experiment, while keeping $T_{Fr}$ at the higher level of −1 °C, $T_{Th}$ is decreased from the default +1 to 0 °C (ML-T-1/0).

In addition, the extreme boundaries of the soil freezing contribution to river discharge are also tested, by setting the $[T_{Fr}, T_{Th}]$ phase change interval unreasonably high at (ML-T10) and also unreasonably low at $[−10.5, −10]$ (ML-T-10). These extreme temperature thresholds allow the soil to remain almost always or never frozen, which consequently should increase or decrease the amount of infiltration to the soil and thus the amount of surface runoff to an extreme level.

**Optimal combination: ML-Opt**

Finally, a prospective experiment is defined with combinations of some evaluated changes that are expected to work best for permafrost. ML-Opt combines incremental changes in ML-Vert, ML-Meta2, ML-Cond2 and ML-T-1 into one experiment. This experiment aims at exploring the interactions among the proposed changes, as feedbacks between different processes can be highly non-linear. For instance, testing the proposed changes in combination can indicate if singular modifications that improve the river discharge simulation, actually (over-) compensate for other sources of errors.

**3. Results**

*3.1. Default Multi-Layer vs. Single-Layer Snow Schemes*

The default MLS generally improves on the SLS, mainly through better bias and variability, with the exception of some parts of the permafrost, where the multi-layer simulation is suboptimal for river discharge.

The impact of the MLS is analyzed first in this section by comparing the default parametrization option ML with the single-layer SL (Figure 3). The ML improves the river flow predictions for the majority of stations over the midlatitudes (about two thirds of them), with a larger cluster of improved catchments (with higher *kge*) present in western/central North America (Figure 3a). The *kge* mean, computed across 453 catchments, is 0.43 for ML while 0.40 for SL, highlighting a small overall improvement. However, in some higher latitude areas, especially in the northern half of Siberia in Asia and also near Alaska in North America (coinciding with the blue box permafrost sensitivity area in Figure 3), the river discharge performance is deteriorated in ML, with many catchments showing a drop of at least 0.05–0.1 (some even above 0.3) in *kge*.

The improvements in midlatitudes seem to come mainly from the smaller bias errors (*abias* decreasing), especially pronounced in central North America (Figure 3c) and to a lesser extent from the smaller variability errors (*avar* decreasing; Figure 3d) and higher correlation (Figure 3b). On the other hand, the deterioration in the higher latitudes over the permafrost seems to relate to much higher variability errors (many catchments with an increase of at least 0.3) and to somewhat higher bias errors and lower correlation in ML.

*3.2. ML Struggles in Permafrost*

Analysis in Siberia demonstrates that the deterioration of the daily river discharge representation and the largely missed snowmelt-driven flood wave is caused by too low surface runoff, which primarily comes from warmer soil in ML, allowing more water infiltrating into the soil and thus reducing surface runoff.

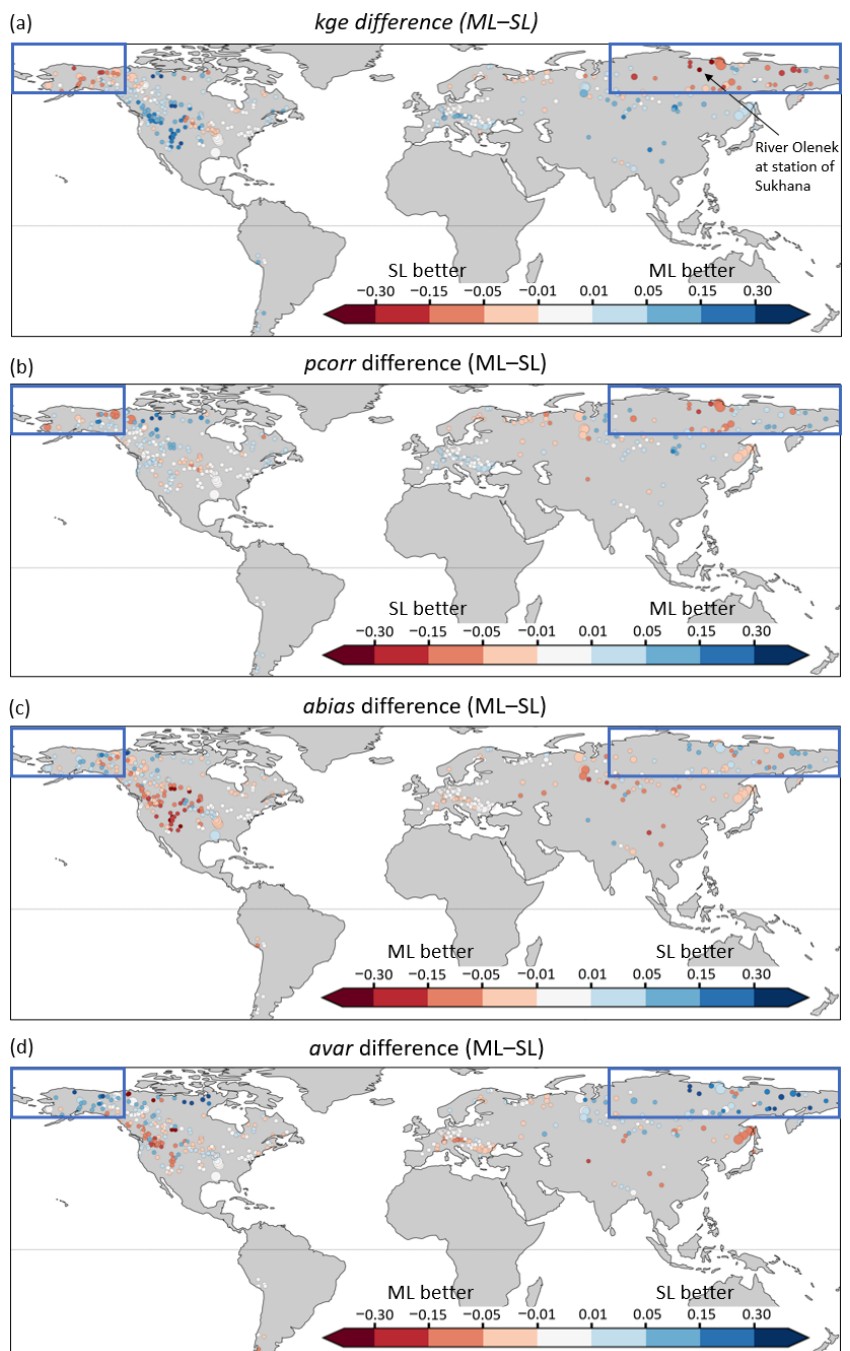

**Figure 3.** Difference of performance metrics between the default multi-layer (ML) and the single-layer (SL) snow scheme experiments, across all 453 stations, calculated on daily river discharge over 1980–2018. (**a**) Modified Kling–Gupta efficiency (*kge*) and (**b**) Pearson correlation (*pcorr*), (**c**) absolute bias ratio (*abias*) and (**d**) absolute variability ratio (*avar*). Improvements in ML are indicated by blue dots in (**a**,**b**), while by red dots in (**c**,**d**). Size of the dots represent the catchment area. The sensitivity area in the permafrost is shown by blue rectangles, while the test catchment on river Olenek at station Sukhana is indicated by black arrow in (**a**).

A test catchment on the Olenek river in Siberia (Sukhana station from within the permafrost sensitivity area; indicated in Figure 3) is used to demonstrate this large negative impact through analyzing the daily climate time series of some key water budget related land-surface variables (snowmelt, surface and subsurface runoff, soil water content and soil temperature) for SL-CDS, SL and ML (Figure 4).

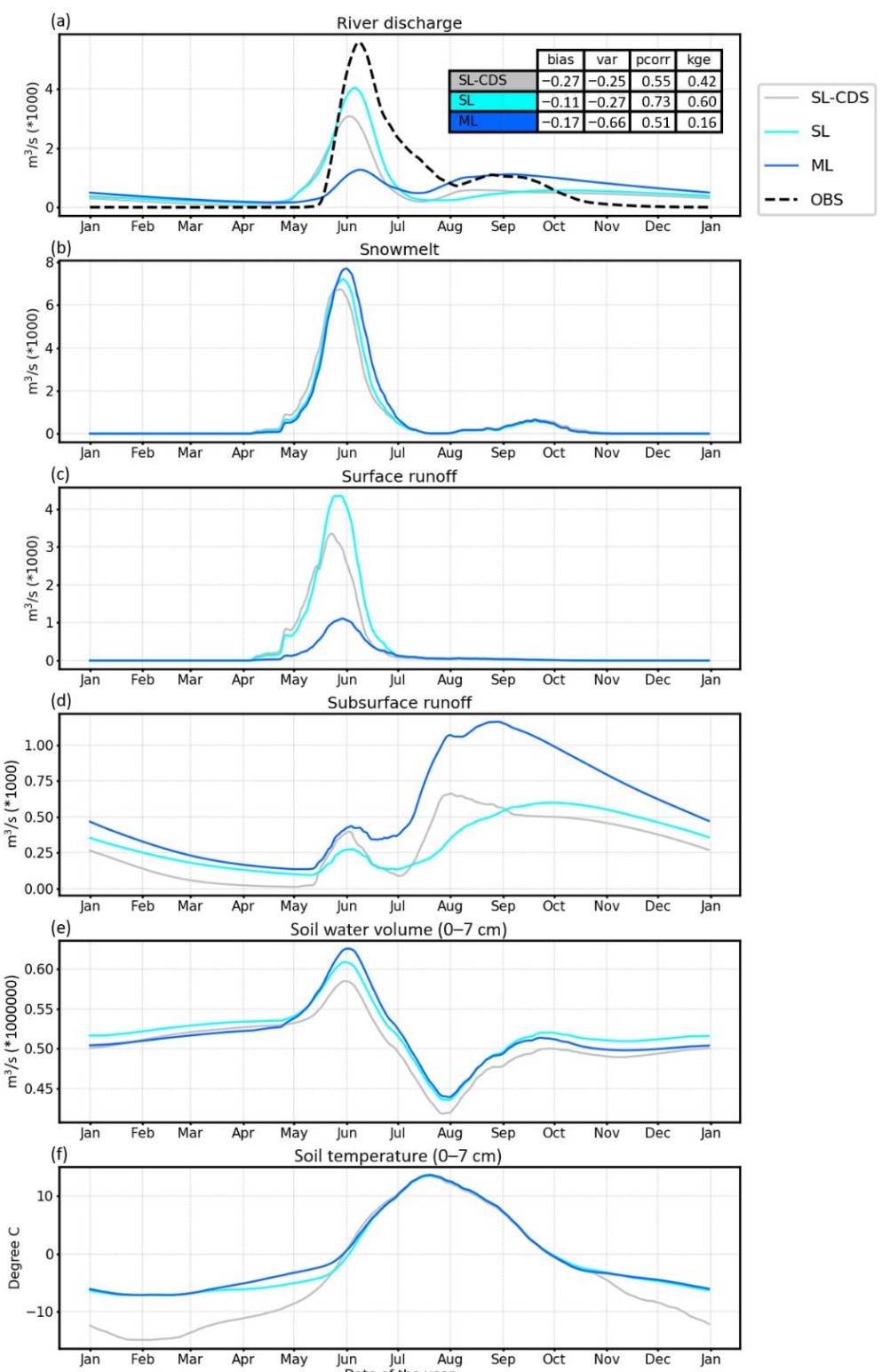

**Figure 4.** Daily climatological mean time series of (**a**) river discharge, (**b**) snowmelt, (**c**) surface runoff, (**d**) subsurface runoff, (**e**) water content and (**f**) temperature in the top 7 cm of the soil from SL-CDS, SL and ML experiments for the Olenek river at the station of Sukhana in eastern Siberia (with area of 127,000 km$^2$). All water related variables are displayed as catchment totals in order to compare them directly to river discharge, while for soil temperature catchment averages are shown (please note the values are divided by 1000 for river discharge, snowmelt and subsurface runoff and by 1,000,000 for soil water volume). The *kge* and its *bias*, *var* and *pcorr* component scores are provided in an inset table for river discharge.

The ML-produced river discharge is clearly inferior compared with both SL and SL-CDS, as it shows a much larger underestimation of the observed flood peak in May–June (Figure 4a). Even though the secondary flood period is better represented by ML in August–September, due to the generally higher river discharge with the exception of May–June. The deterioration of the flow is reflected in the ML scores being the lowest across all three experiments (inset table in Figure 4a). The only exception is the bias ratio which is lowest for SL-CDS. This very low negative bias is a consequence of the snow data assimilation that removes water from the rivers in high latitudes. It is related to the current single layer snow model's tendency to melt the snow too slowly, which is compensated by the assimilation system during snowmelt periods, as documented in Zsoter et al. [39].

The snowmelt peak in May–June is slightly higher and delayed in ML, compared with SL and even more so with SL-CDS (Figure 4b). At the same time, the soil is better insulated by the multi-layer snow in ML during March–May, as the temperature is higher by up to 2 °C than in SL (Figure 4f). As the snow is better insulated in ML, the melting will start later during spring, which then delays the faster soil temperature increase in ML, occurring in end of May. The surface runoff is highest in SL and lowest in ML during April–June, also highlighting a similar delay seen on the snowmelt peaks (Figure 4c).

The higher soil temperature in ML implies that a smaller fraction of the water is frozen in the soil, compared to SL, during the crucial snowmelt period and more water can infiltrate into the soil, which then reduces the amount of water that can runoff directly. The higher infiltration will increase the water content in the soil during May–July (Figure 4e), which will result in increased subsurface runoff (Figure 4d). This happens with a delay, as the water needs to reach the bottom of the soil to leave as subsurface runoff, producing a delayed peak to around end of July–September (Figure 4d).

The missing river discharge in ML is clearly related to the too low surface runoff during the April–June snowmelt, which period is clearly better represented in both SL and SL-CDS. The small differences in snowmelt cannot explain the lower surface runoff amounts. If anything, it should likely contribute to an increase in ML. In addition, the deficit in surface runoff is offset by the higher subsurface runoff in ML. However, the excess water from subsurface runoff spreads out over most parts of the year and thus cannot compensate for the missing surface runoff during the snowmelt flood period. Instead, the crucial aspect of the changes in ML is the higher soil temperature. This soil temperature difference is most critical in the middle of May, when the snowmelt is rapidly increasing in the catchment.

*3.3. Improving the Multi-Layer Snow Scheme Performance in Permafrost*

In this section, the ECLand sensitivity experiments (Figure 2 and Section 2.8) are evaluated for their ability to improve the river discharge simulation, focusing on permafrost.

3.3.1. Impact of the ECLand Experiments on a Test Catchment in Siberia

The ECLand permafrost sensitivity experiments are demonstrated to increase the surface runoff in a test catchment in permafrost, by primarily making the soil colder through a series of incremental changes in the snow and soil freezing parametrizations, including modifications of the snow vertical discretization, snow density metamorphism, snow-soil thermal conductivity and soil freeze temperature.

In the Sukhana test catchment, the modifications are clearly effective in altering the amount of water distributed amongst different parts of the water budget (Figure 5). Generally, the impact on snowmelt (Figure 5b) is the smallest (maximum of a few days delay and a slight change in magnitude), while all other variables show much larger variability between the experiments (Figure 5a,c–f).

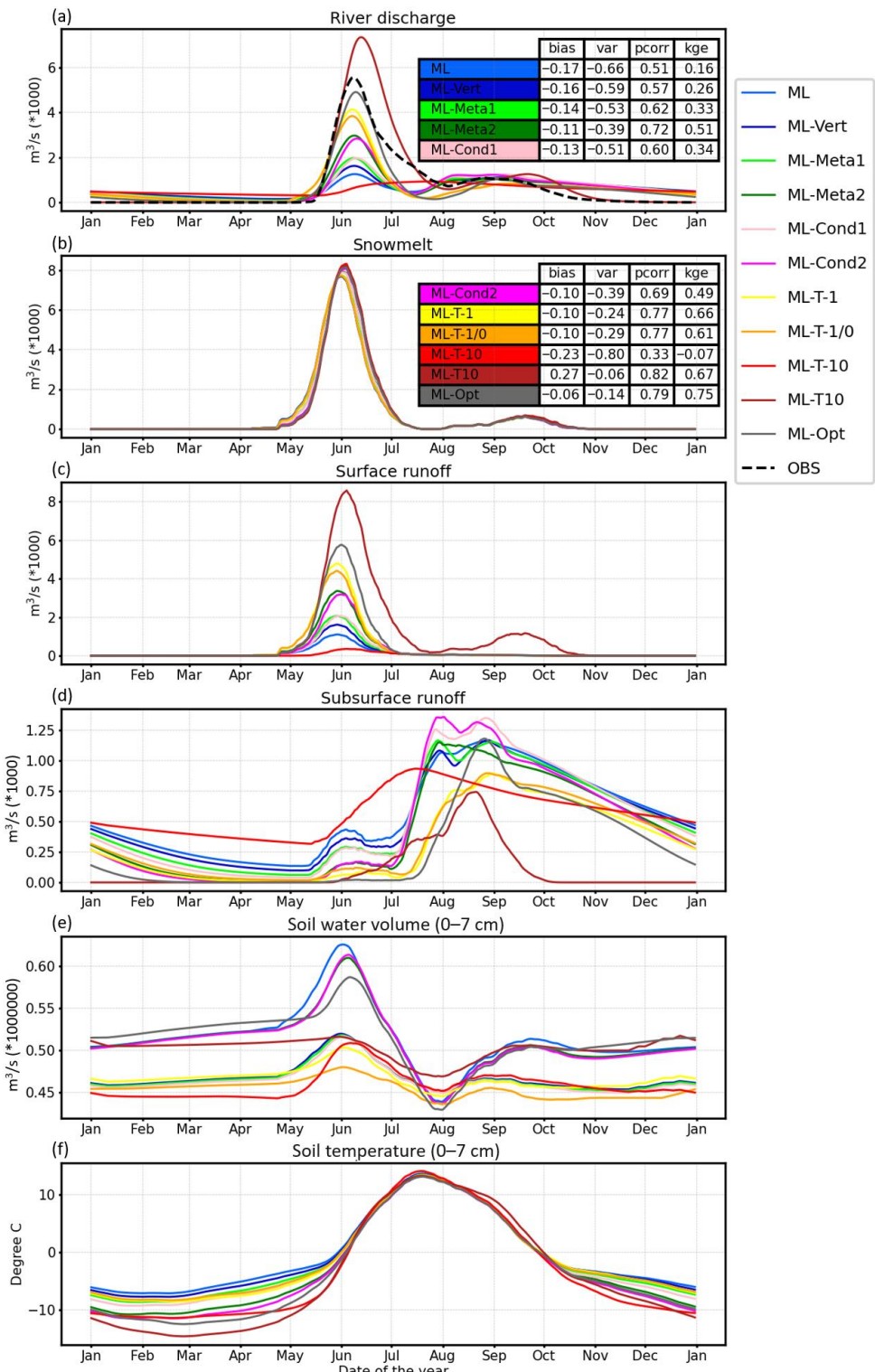

**Figure 5.** Daily climatological mean time series of (**a**) river discharge, (**b**) snowmelt, (**c**) surface runoff, (**d**) subsurface runoff, (**e**) water content and (**f**) temperature in the top 7 cm of the soil from ML, ML-Vert, ML-Meta1, ML-Meta2, ML-Cond1, ML-Cond2, ML-T-1, ML-T-1/0, ML-T-10, ML-T10 and ML-Opt experiments for the Olenek river at the station of Sukhana in eastern Siberia (with area of 127,000 km$^2$). All water related variables are displayed as catchment totals in order to compare them directly to river discharge, while for soil temperature catchment averages are shown (please note the values are divided by 1000 for river discharge, snowmelt and subsurface runoff and by 1,000,000 for soil water volume). The *kge* and its *bias*, *var* and *pcorr* component scores are provided in an inset table for river discharge.

The surface runoff shows very large sensitivity to the ECLand changes in May–June, coinciding with the main snowmelt period (Figure 5c). The river discharge behavior (Figure 5a) is directly determined by the variability in surface runoff in this period. The vertical discretization adjustment (ML-Vert), which is active for about 25% of the test catchment area, usually during October–May, and the combination with the first snow metamorphism change (ML-Meta1) both add small increase. However, ML-Vert combined with the second snow metamorphism setting (ML-Meta2) produces a much larger impact on surface runoff. On the other hand, the change in surface conductivity does not appear to be effective as both ML-Meta1 with ML-Cond1 and ML-Meta2 with ML-Cond2 (both pairs sharing the same snow metamorphism change respectively) show similar levels of surface runoff.

The experiments with the largest impact on surface runoff (and thus river discharge) are those including changes of the soil freezing. The experiments of ML-T-1 and ML-T-1/0 help reaching the surface runoff level of SL by resetting the freeze temperature to −1 °C and the thaw temperature to 0 °C. ML-Opt shows even further improvement with a surface runoff peak that closely matches the shape and magnitude of the observed river discharge until middle of June. This is achieved by the combined impact of the vertical discretization, second snow metamorphism, surface conductivity and freeze temperature changes. On the extreme end of the spectrum, the ML-T10 experiment shows too high surface runoff, by setting the freeze and thaw temperatures to an unrealistically high level (at around +10 °C), consequently partitioning much of the runoff into surface runoff.

The soil temperature changes are in agreement with the surface runoff behavior described above. The soil is warmest in ML, while it gradually gets colder by the ECLand modifications (Figure 5f). The soil is coldest for ML-Opt and the two extreme temperature experiments, ML-T-10 and ML-T10, during the main snowmelt season in March–May. The cooling soil means, the temperature will be more and more likely closer or below the freeze temperature threshold ($T_{Fr}$), which progressively decreases the infiltration into the soil. This leads to generally reduced soil water content (Figure 5e) and decreased subsurface runoff (Figure 5d), although with large variability depending on the actual parametrization changes and the time of year.

The *kge* and the three component scores (inset tables in Figure 5a,b) confirm the gradually improving behavior of these experiments from ML to ML-Opt, through the introduction of the incremental ECLand parametrization changes. The *bias* improves from −0.17 to −0.06, *var* from −0.66 to −0.17, *pcorr* from 0.51 to 0.71 and finally the *kge* from 0.16 to 0.75, representing a very large jump in skill.

### 3.3.2. Impact of the ECLand Experiments in Permafrost

The series of ECLand experiments are shown to provide widespread improvement for river discharge in the permafrost sensitivity area, based on all available catchments. They result mainly in reduced bias and variability errors, with optimal performance achieved by the ML-Opt experiment, including the combined changes of vertical discretization, second snow metamorphism, surface conductivity and freeze temperature.

The 69 catchments, found in the sensitivity area defined in the 60–80 N latitude belt over the permafrost (Figure 1), were collectively analyzed to see if the conclusions obtained from the test catchment in Siberia can be generalized to the permafrost. Figure 6 provides a visual summary of the overall performance in simulating river discharge, averaged over the sensitivity area, where the default configuration ML performs unfavorably. Each experiment is represented by a dot in a 3-dimensional graph showing *bias* (*x*-axis), *var* (*y*-axis) and *pcorr* (symbol size).

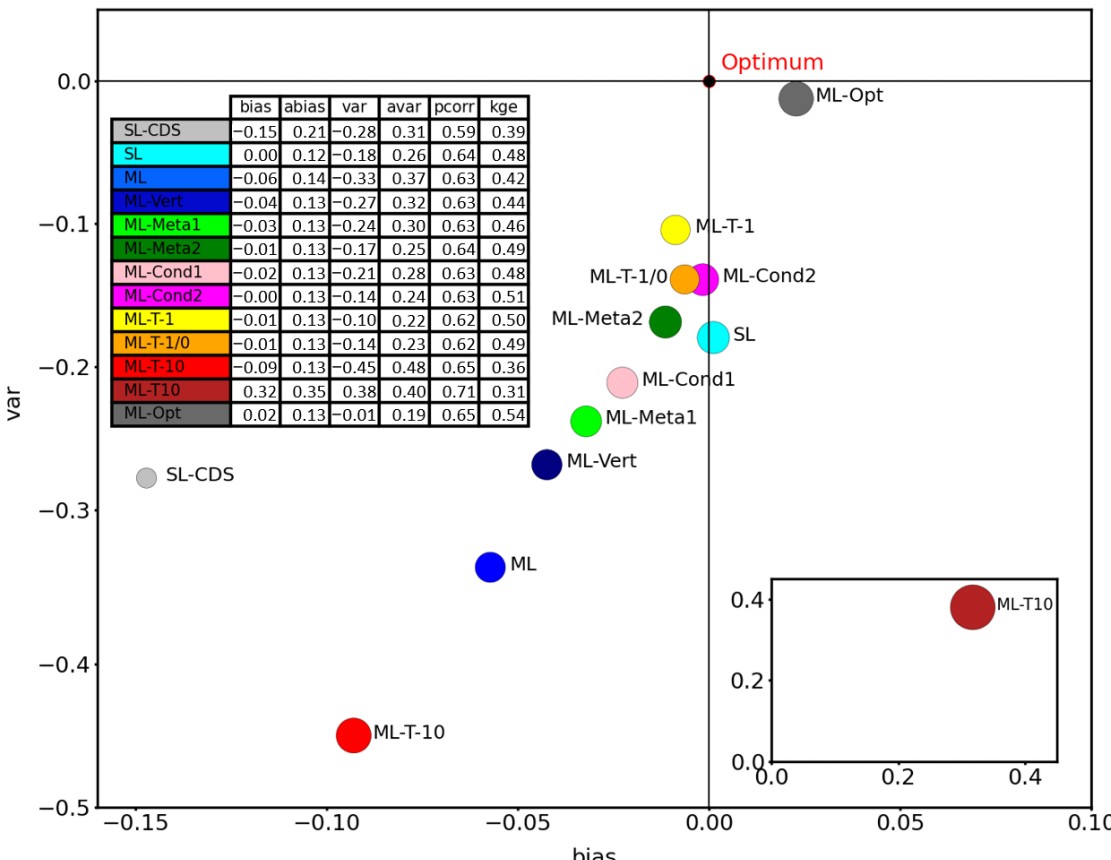

**Figure 6.** Scatter plot of area average scores for the 13 analyzed experiments. The scores are computed from 69 catchments in the 60–80 N belt of the permafrost (see Figure 1 for the area). The size of the dots represents *pcorr*. The outlier ML-T10 is displayed in an inset graph for better readability. The area average score values are provided for all analyzed experiments in the inset table.

The ECLand parametrization changes are effective to improve the simulation of river flow in the permafrost area, in particular the bias and variability errors, which improve from ML-T-10 to ML-Opt (see also the average scores of 69 catchments in an inset table in Figure 6). The correlation is less impacted by the modifications, the difference between all experiments is mostly within a few percent (the two extremes have average correlation of 0.59 (SL-CDS) and 0.71 (ML-T10)).

The furthest from the optimum [0; 0] *bias/var* point are the ML-T-10 and ML-T10 extreme temperature experiments. Nevertheless, the highest average correlation is achieved with the positive extreme, when the soil remains mostly frozen. The fact that an unrealistically extreme-setup simulation, with excess surface runoff, produces the highest correlation shows that the current land-surface process representation in ECLand is still suboptimal in the permafrost, even after the parametrization changes. Another outlier is the online produced SL-CDS, which has lower scores, especially the very low negative *bias*, due to the snow data assimilation removing water from the rivers in the northern latitudes, as a compensation for the slow snowmelt in SLS [39].

Overall, there is a notable improvement achieved by the ECLand experiments, shown by the increase in *kge* from 0.42 (ML) to 0.54 (ML-Opt), while the single-layer experiments highlight lower *kge* of 0.48 (SL) and 0.39 (SL-CDS).

### 3.3.3. Global Impact of the ECLand Experiments

The changes in the snow and soil parametrization schemes in ECLand could improve the river discharge in the permafrost sensitivity area in eastern Siberia and Alaska. However,

in other areas these changes are dominantly suboptimal, as the difference between ML-Opt, the highest performing ECLand experiment, and ML highlights (Figure 7).

| | bias | abias | var | avar | pcorr | kge |
|---|---|---|---|---|---|---|
| SL-CDS | −0.02 | 0.27 | −0.14 | 0.25 | 0.63 | 0.42 |
| SL | 0.23 | 0.30 | −0.05 | 0.25 | 0.67 | 0.40 |
| ML | 0.18 | 0.28 | −0.11 | 0.25 | 0.68 | 0.43 |
| ML-Vert | 0.19 | 0.29 | −0.08 | 0.25 | 0.68 | 0.42 |
| ML-Meta1 | 0.19 | 0.29 | −0.08 | 0.24 | 0.67 | 0.42 |
| ML-Meta2 | 0.20 | 0.29 | −0.05 | 0.24 | 0.67 | 0.42 |
| ML-Cond1 | 0.20 | 0.29 | −0.07 | 0.24 | 0.68 | 0.42 |
| ML-Cond2 | 0.21 | 0.29 | −0.04 | 0.23 | 0.67 | 0.42 |
| ML-T-1 | 0.21 | 0.30 | −0.04 | 0.23 | 0.66 | 0.41 |
| ML-T-1/0 | 0.23 | 0.31 | −0.04 | 0.24 | 0.66 | 0.39 |
| ML-T-10 | 0.15 | 0.28 | −0.12 | 0.28 | 0.69 | 0.41 |
| ML-T10 | 1.39 | 1.41 | 0.43 | 0.48 | 0.59 | −0.76 |
| ML-Opt | 0.23 | 0.31 | −0.00 | 0.22 | 0.66 | 0.41 |

**Figure 7.** Difference of the modified Kling–Gupta efficiency (*kge*) between ML-Opt and ML. Blue areas show better performance in ML-Opt, while red indicates better performance in ML. The sensitivity area in the permafrost is shown by blue rectangles, while the test catchment on river Olenek at station Sukhana is indicated by black arrow. The area average scores are provided for all analyzed experiments in the inset table.

The worst ML-Opt performance is found in the western part of USA, Canada and catchments in central Asia, directly in those areas where ML could improve the most on SL (see Figure 2). Interestingly, the Canadian Arctic regions (Nunavut) do not show similarly improving results to other permafrost regions. One source of error, which can contribute to this, is the underlaying physiographic data used to drive the simulations, e.g., lake cover. Boussetta et al. [21] showed that recent lake cover datasets are substantially different from those used in this work for the Nunavut region. The use of more realistic physiographic data will be explored in future work. Furthermore, unrepresented physical processes in the land-surface model can affect the hydrological cycle of these regions, e.g., wind-driven sublimation of the snowpack which can change the amount of snow mass available for melting at the end of the season. Similarly, the handling of frozen lakes with, e.g., snow not accumulating on top of lake ice in ECLand, may introduce compensating errors with the changes tested in ML-Opt.

Nevertheless, even though only about one third of the stations show improvement by ML-Opt, the globally averaged scores show that ML is only slightly better (inset table in Figure 7). Moreover, the score variability amongst all the experiments, in general, is quite small as well. The only real exception is ML-T10 that has very low scores throughout. The geographical maps of the *kge* differences between the experiments (from ML-Vert to ML-T10) and ML is provided in the Supplementary Materials (Figures S1–S9).

The deterioration of ML-Opt and the other ECLand experiments in milder climate areas is related to the surface runoff increase during the snowmelt period, demonstrated earlier. Especially in catchments where the *bias* in ML (and similarly in SL) was originally positive, e.g., around the Rockies (as suggested by Zsoter et al. [39]), any further increase in snowmelt-related surface runoff will dominantly be detrimental and contribute to further increased *bias* and thus lower *kge*.

## 4. Discussion

It was shown in this study that the MLS has an improved representation of the hydrology, with the notable exception of the coldest areas in permafrost, where the SLS is

superior. To improve the MLS hydrological representation in permafrost, modifications of the ECLand snow and soil freezing schemes were tested. It was shown that a series of incremental changes could noticeably improve the quality of the river discharge simulation over a large area in permafrost, primarily through decreasing the soil temperature and thus increasing the amount of surface runoff in the critical spring snowmelt period.

**Improving the hydrological process representation**

The results have demonstrated that the use of uniform parameters in ECLand in the snow and soil freezing schemes, currently applied in ECLand, are too simplistic and will not work for both the permafrost and non-permafrost areas in the snow-impacted climate. Spatially variable parametrization for variables, such as $c_\zeta$ in the destructive metamorphism of the snow, $l_b$ in the snow-soil thermal conductivity and $T_{Fr}$ freeze temperature in the soil freezing scheme, explored in this work, promise to bring a more balanced approach for delivering improved hydrological process representation.

However, this could technically be complex to implement in ECLand and needs a substantial amount of further research. As a first step, the parametrization part of ECLand has recently been refactored (by removing hard-coded parameter values) and the Multiscale Parameter Regionalization (MPR; [52]) has been implemented for estimation of spatially varying parameters. This change will make it easier in the future to work on aspects of ECLand, such as the one explored in this study for the snow and soil parametrization impact on hydrology.

This study has also contributed to the understanding of the hydrological importance of each tested change in the ECLand parametrization, which could be based on the average verification metrics in the permafrost (distance between dots in Figure 6), supported by the land-surface processes representation at the example catchment (Figure 5).

The thaw temperature adjustment from +1 to 0 brought deterioration, while all other changes could improve river discharge. The smallest improvement seems to come from the snow-soil thermal conductivity adjustment (updated $l_b$), closely followed by the first snow metamorphism change (variable $c_\zeta$). The snow vertical discretization change looks to bring larger improvements (adjustment over complex terrain), while the second snow metamorphism change (uniform $c_\zeta$) is even more beneficial in the permafrost. Finally, the biggest contribution appears to come from the adjustment of the lower temperature threshold in the soil freezing scheme (freeze temperature) from $-3$ to $-1$ °C.

**Land-surface modelling challenges**

Due to the presence of a unique soil column within a grid cell in EC-Land, the soil temperature may be largely affected by the repartitioning of the sub-grid tiled surfaces, as these can be characterized by different surface energy fluxes. In high latitude during transition seasons (e.g., snow accumulation and ablation), the tile fraction subdivision depends on the snow cover fraction, exerting a large control on the heat flux conducted through the soil ([19,53,54]). When the snow cover fraction is less than one, part of the soil directly interacts with the atmosphere above, without the thermal insulation effect of the snowpack. This in turn, may affect the soil temperature and therefore the fraction of frozen water within the soil. Future modelling work should evaluate the hydrological sensitivity to snow cover fraction parametrization in global land-surface simulations (see for example [55]), considering the combined effect that this can have on the soil below and the atmosphere above.

In addition, future modelling work should also evaluate the effect of unrepresented or poorly represented hydrological processes in ECLand. These include, for example wind-driven snow sublimation, snow interception by forests, as well as of a more physically-complex representation of frozen/unfrozen water phases in the soil column, which all promise to bring benefits to the hydrological cycle, especially in permafrost regions (see for instance [10,11,44]. Moreover, the impact of a deeper soil column, with additional vertical layers, has also been shown to work better for permafrost simulation in land-surface models ([56]). More complex models for hydrological applications in high-latitude regions

have been proposed in the literature (e.g., [44,56,57]). Their implementation in ECLand could be considered in the future.

**Relevance for ECMWF**

The next operational IFS cycle (48r1) of ECMWF will include multi-layer snow representation. Based on this study, the slightly modified version of ML-Meta1, was selected for operationalization, which includes the snow vertical discretization and first snow metamorphism changes on top of the default ML configuration. The results of this study highlighted that this new snow model did show moderate hydrological improvements in permafrost and could still retain most of the good performance of the default MLS in other areas.

**Earth system modelling implications**

Modelling improvements in the Earth system process representation, such as the use of any prospective MLS version explored in this study, require thorough testing in coupled forecast mode with data assimilation, similarly as in [26]. This is necessary to check the transferability of the offline-demonstrated hydrological improvements, and make sure other variables, such as 2-metre temperature and surface fluxes of sensible and latent heat, will not deteriorate.

This study has proven that further development of the snow and soil parametrizations in ECLand is crucial to achieve better hydrological performance everywhere globally; however, it could not be part of this study and will only be explored in the future as an important research area.

The success of helping the operational development of the ECLand model's snow component has shown that hydrological studies, such as the work presented in this paper, have great potential to help improve the land-surface realism in Earth system models and can contribute to improvements in not just the hydrological variables, such as river discharge, but potentially other components of the Earth system as well.

**Limitations of the study**

The authors acknowledge that even with the best care taken in the experimental setup of this study, some limitations remain. The network of analyzed catchments is still under-representative in some areas, also with the relatively short minimum observation length of eight years as a compromise. Similarly, this study was not designed to be a full sensitivity experiment for the land-surface processes in the permafrost, which would have required a much higher computational cost. In addition, the choice of testing ECLand changes in MLS was because of practical reasons, but most of the modifications would be expected to show improvements in a similar manner even if tested within the SLS.

**5. Conclusions**

The representation of snow is a crucial aspect of land-surface modelling, as it has a strong influence on the energy and water balances. Snow schemes with multiple layers can better represent the snowpack evolution and bring improvements on single-layer schemes in simulating the snow processes and contribute to better soil freezing and hydrological cycle.

In this paper, the hydrological impact of the MLS, implemented in ECLand, was analyzed globally with ERA5-forced experiments over the period of 1979–2018. The CaMa-Flood model was used to generate river discharge from the ECLand runoff output, allowing direct comparison with gauged observations over more than 400 snow-impacted catchments. Different sensitivity experiments were conducted to evaluate the impact of changes in the ECLand snow and soil freezing schemes on the terrestrial hydrological processes, with particular focus on permafrost.

It was found that while the default MLS generally improves the river discharge simulation, mainly through better bias and variability errors, the performance is suboptimal in large parts of the high latitude permafrost regions. The analysis of the climatological mean time series, in a test catchment in Siberia, demonstrated that the largely underestimated

snowmelt-driven floods in late spring to early summer are caused by too low surface runoff. The MLS better insulates the soil, which allows more water to infiltrate into the soil and thus surface runoff is reduced.

It was also found that the ECLand experiments provide widespread improvement for river discharge in the sensitivity area, defined in permafrost. The incremental changes of the snow vertical discretization, destructive metamorphism, snow-soil thermal conductivity and soil freeze temperature lead to gradually colder soil, which resulted in increased surface runoff and thus better river discharge simulation during the critical snowmelt-driven flood period. The ML-Opt experiment, as the combination of the best performing ECLand changes, has shown higher overall *kge*, mainly through reduced bias and variability errors.

The results, presented here, have directly influenced the MLS version that will be introduced in the next Integrated Forecasting System cycle of ECMWF (48r1). This has demonstrated that hydrological analyses, such as the work presented in this paper, can provide a useful platform to diagnose areas where improvements are needed in the land-surface representation of the Earth system models.

**Supplementary Materials:** The following supporting information can be downloaded at: https://www.mdpi.com/article/10.3390/atmos13050727/s1, Figure S1: Difference of the Kling–Gupta efficiency (*kge*) between ML-Vert, the experiment with modified snow vertical discretisation over complex terrain with at least 25 cm snow depth, and ML, the default multi-layer experiment, across all 453 stations, calculated on daily river discharge over 1980–2018. Improvements in ML-Vert are indicated by blue dots. Size of the dots represent the catchment area; Figure S2: As Figure S1, but *kge* difference between ML-Meta1 (the 1st modification of the destructive metamorphism of the snow with variable $c_\xi$ parameter values across the snow layers, together with ML-Vert) and ML; Figure S3: As Figure S1, but *kge* difference between ML-Meta2 (the 2nd modification of the destructive metamorphism of the snow with one $c_\xi$ parameter value as in Anderson et al., 1976, together with ML-Vert) and ML; Figure S4: As Figure S1, but *kge* difference between ML-Cond1 (ML-Meta1 together with the snow-soil thermal conductivity computation with the revised parameter $l_b = 0.5$) and ML; Figure S5: As Figure S1, but *kge* difference between ML-Cond2 (ML-Meta2 together with the snow-soil thermal conductivity computation with the revised parameter $l_b = 0.5$) and ML; Figure S6: As Figure S1, but *kge* difference between ML-T-1 (change of the soil freeze parameter $T_{Fr}$ from the default $-3$ to $-1$ °C added onto ML-Meta1) and ML; Figure S7: As Figure S1, but *kge* difference between ML-T-1/0 (change of the freeze temperature $T_{Fr}$ to $-1$ °C and thaw temperature $T_{Th}$ from the default $+1$ to $0$ °C, added onto ML-Meta1) and ML; Figure S8: As Figure S1, but *kge* difference between ML-T-10 ($[T_{Fr}, T_{Th}]$ changed to $[-10.5, -10]$, added onto ML-Meta1) and ML; Figure S9: As Figure S1, but *kge* difference between ML-T10 ($[T_{Fr}, T_{Th}]$ changed to $[+10, +10.5]$, added onto ML-Meta1) and ML.

**Author Contributions:** Conceptualization, E.Z., G.A., C.P., E.S. and H.C.; Data curation, E.Z. and G.A.; Methodology, E.Z., G.A., C.P., E.S. and H.C.; Visualization, E.Z.; Writing—original draft, E.Z.; Writing—review and editing, E.Z., G.A., C.P., E.S. and H.C. All authors have read and agreed to the published version of the manuscript.

**Funding:** Ervin Zsoter's PhD is supported by the Wilkie Calvert Co-Supported PhD Studentships at the University of Reading. Christel Prudhomme and Ervin Zsoter were supported by the Copernicus Emergency Management Service—Early Warning Systems (CEMS-EWS). Hannah Cloke is supported by the EVOFLOOD project: The Evolution of Global Flood Risk UK NERC, NE/S015590/1. Elisabeth Stephens and Hannah Cloke are supported by the FATHUM project: Forecasts for AnTicipatory HUManitarian Action funded by UK NERC as part of their Science for Humanitarian Emergencies & Resilience (SHEAR) program, NE/P000525/1.

**Institutional Review Board Statement:** Not applicable.

**Informed Consent Statement:** Not applicable.

**Data Availability Statement:** Not applicable.

**Acknowledgments:** We are grateful to The Global Runoff Data Centre, 56068 Koblenz, Germany for providing observations for our river discharge analysis.

**Conflicts of Interest:** The authors declare no conflict of interest. The funders had no role in the design of the study; in the collection, analyses, or interpretation of data; in the writing of the manuscript, or in the decision to publish the results.

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
