# Peer review of "Hydrological Impact of the New ECMWF Multi-Layer Snow Scheme"

_atmosphere, doi:10.3390/atmos13050727_

Round 1

Reviewer 1 Report

The authors discussed an important topic of simulating the snow cover and its impact on the modeling of surface runoff. The article was prepared correctly. It is based on the current literature. The research methodology is clearly described. The results are presented clearly, and the visualization of the results in the figures is fully legible and facilitates their interpretation. I have no comments on the work, I can only suggest a slight shortening of the abstract, and at the same time a greater focus on the reasons for conducting the research presented in the article.

Author Response

Dear reviewer, thank you very much for your review. Please find attached our responses.

Reviewer 2 Report

This paper addresses the interesting issue of the role of snow in hydrological processes. However, I note some shortcomings that need to be addressed.

 I consider as inappropriate the exclusion of catchments where the visible influence of dams and lakes was found. In light of the increasing frequency of extreme hydrological events, increasing attention is being paid to hydrological development and flow regulation. Making a preliminary assumption creates a purely theoretical work with no applied possibilities.

What is the substantive justification for choosing a catchment area of at least 5000 km2? The dynamics of runoff due to snow recharge will be greater in small catchments especially in mountainous areas.

I can't find any data on the land use structure of the catchment areas selected for the analysis.

Moreover, there is no basic information on the geological structure of the catchment area, i.e. larger or smaller water infiltration capacities.

Author Response

Dear reviewer, thank you very much for the review. Please find attached our responses.

Reviewer 3 Report

Review of the paper “Hydrological impact of the new ECMWF multi-layer snow scheme” submitted to Atmosphere by Zsoter et al.

In this paper, Zsoter et al. evaluate the impact for hydrological modelling of a multi-layer snowpack scheme recently implemented in the ECLand surface model used at ECMWF. The hydrological evaluation is carried out at a global scale for catchments located in cold regions where snow influences the hydrological cycle. The atmospheric data driving EC Land are taken from the ERA5 global reanalysis over the period 1979-2018. Results of hydrological simulations obtained with the multi-layer snowpack scheme are first compared with simulations obtained with the single layer snowpack scheme available in EC-Land. The authors show that hydrological simulations are generally improved with the multi-layer snowpack scheme compared to the single layer scheme, except in regions where the hydrological cycle is influenced by the presence of permafrost. To improve the simulations in these regions, the authors propose additional simulations that test the sensitivity of the hydrological simulations to the (i) snow density evolution and its impact on snow thermal conductivity, (ii) snow layering (iii) soil/snow coupling and (iv) soil freezing scheme. The authors managed to develop an optimized version of the model that greatly improves the hydrological simulations in permafrost regions due an improved partitioning between surface runoff and infiltration. However, this optimized configuration for permafrost regions slightly decrease model performances in non-permafrost regions compared to the default simulations using the multi-layer snowpack scheme, illustrating the need for further development of the EC Land model to better represent cold-region hydrology.

The subject of this paper is very relevant for the hydrological community interested in hydrology at continental scale and in cold regions. This paper illustrates well the challenges that are facing land surface schemes initially developed in a NWP context to simulate the hydrological cycle in cold regions. However, this paper lacks a clear quantification (and/or a detailed discussion) of the impact on hydrological simulations of the snow cover fraction that can affect greatly the atmosphere/snow/soil system and the dynamics of soil freezing and thawing in Northern latitudes. In addition, this paper would benefit for (i) a more detailed assessment of some of the changes proposed to the snowpack scheme and (ii) a careful analysis of the impact of initial condition of land surface variables on the hydrological simulations. For these reasons, I recommend major revisions of this paper. My main comments are listed below as general comments and are followed by specific and technical comments.

General Comments

1. In Section 2.8, the authors propose a large set of experiments to test the impact of the snow and soil freezing schemes on hydrological simulations in permafrost regions. These tests mostly concern the thermal properties of the snow cover (that depends on the snow density), its coupling with the underlying ground and the soil freezing dynamics in the soil scheme. The authors never discuss the fact that the EC Land scheme is using a unique soil column to represent the evolution of soil temperature and moisture over a given grid cell. The surface itself is divided into different tiles that interact with the atmosphere and this unique soil column. Therefore, the snow cover fraction plays a key role in modulating the exchanges of energy between the atmosphere and the soil and affects the evolution of soil temperature. When the snow cover fraction is lower than 1, part of the surface and the soil below directly interacts with the atmosphere which may lead to intense cooling of the soil in early winter when the snowpack is shallow and does not cover the whole grid cell. Therefore, the authors should present in their paper a detailed description of the snow cover parameterization used in EC Land and quantify its impact on the soil freezing dynamics. The snow cover fraction in land surface models has often been developed to compute the surface albedo and its formulation is highly empirical and may not be suitable when considering the energy exchanges between the atmosphere, the snow and the soil.

To test the impact of this parameterization, the authors could add two new simulations in their paper (using the default ML configuration). In the first test, they could assume that the snow cover fraction reaches a value of 1 as soon as snow is present on a grid cell. Such approach is often used in point scale land surface simulations (e.g. Brun et al., 2013). In the context of this paper, it could be considered as an extreme test following the same philosophy as the ML-T10 and ML-T-10 tests. In a second configuration, they could a different formulation for the snow cover fraction (e.g. Niu and Yang, 2007) and assess the impact on hydrological simulations. In the present version of the paper, it is difficult to estimate if some if the changes proposed in the snow scheme (e.g. ML-META1 and ML-META2) compensate for some limitations in the snow cover parameterization (see my next general comment). Adding the tests related to the snow cover fraction would be very beneficial for the community using land surface models in support in hydrological forecasting. Indeed, all these models use different formulations for the snow cover fraction and their effects on the soil thermal dynamics and the hydrological response are not properly documented in papers.

In addition to the description of the snow cover fraction in EC land, the authors should also describe how EC Land simulates the evolution of the snow cover over the two surface tiles covered by snow: exposed and shaded snow (Boussetta et al., 2020). Is the model using the same formulation for the snow cover fraction for the two types of snow cover? How are treated the snow/canopy interactions? This representation could have a large impact of the hydrological simulations since the boreal forest partially covers many catchment considered in this study.

2. The snow density evolution in the multi layer snowpack of EC Land accounts for destructive metamorphism to represent the increase in snow density of fresh snow following the formulation proposed by Anderson (1976). As reported by Cao et al. (2020), it seems that the formulation implemented in the EC Land snow schemes includes an error of the parameter $c$ resulting from an error in unit conversion from the original formulation of Anderson (1976) in cm3 g-1 to m3 kg-1. Therefore, the authors proposed two tests where the values of the parameter $c$ are adjusted. In particular, the experiment ML-Meta2 uses the original value proposed by Anderson (1976). This modification led to lower soil temperature and improved the simulations of the hydrological response in springtime in the region covered by permafrost. It would be very useful for the reader to know if the hydrological simulations are improved for the good reasons and if the properties of the simulated snow cover (especially density and snow depth) are really improved in the configurations ML-Meta1 and ML-Meta2. The authors could for example use the ESM-SnowMIP database to evaluate the impact of the configurations ML-Meta1 and ML-Meta2 on simulated snow density. It could be done with offline point scale simulations driven by observed forcing (as done in Boussetta et al. (2020) when evaluating the ML snow scheme and the ML Vert configuration).

Overall, does the configuration ML-Meta2 solve the underestimation of Arctic snow density reported in Cao et al. (2020)? Over the last 5 years, several publications have discussed the limitations of multi-layer snowpack models to simulate the vertical profile of density in Arctic snow and its impact on the permafrost thermal regime (Barrere et al., 2017; Gouttevin et al., 2018; Domine et al., 2019; Royer et al., 2021). These papers have proposed different modelling approaches and techniques to overcome this limitation that combine an improved representation of the effect of wind on near surface snow density (via compaction during blowing snow events) and a limitation of compaction for the deeper snow layers in presence of low vegetation at the bottom of the snowpack. It would interesting to discuss the results of the configurations ML-Meta1 and ML-Meta2 in the context of these papers. I understand that improving hydrological simulations in permafrost environments is the main objective of these modifications but to my point of view it is crucial for the community to understand if these modifications goes in the right direction for the right reasons.

3. Based on the sentence at L 279-281, I understand that the EC Land offline simulations and the corresponding CaMa-Flodd simulations were carried out over the period 1979-2018 and the results were analyzed over the period 1980-2018. Therefore, it seems that authors used a 1-year spin-up for the simulations. This spin-up duration is small in the context of offline land surface modelling to provide realistic initial soil moisture and temperature profile across the simulation domain, in particular in permafrost regions. Brun et al. (2013) used for example a 14-yr spin-up period when simulating the evolution of snow and soil properties across Northern Eurasia with a multi-layer snowpack scheme coupled to a multi-layer soil scheme. Recently, Elshamy et al. (2020) have explored the impact of different initialization approaches on permafrost simulations with a large-scale hydrological model in Canada. They have shown that a 1-year spin-up is not sufficient to get realistic initial conditions in permafrost regions and recommended to repeat multiple times the 1-year spin up period to reach soil temperatures consistent with the local climate conditions. Therefore, it would be very interesting for the reader if the authors could add more details about the generation of initial conditions for the surface for their 13 offline simulations. I also recommend the authors to investigate if the error metrics for the hydrological simulations change when a 30-year period (1988-2018) is considered for the evaluation instead of the full period (1980-2018), in particular for stations located in the permafrost region.

Specific Comments

Abstract L 27: the terminology “snow density metamorphism” is not really clear and should reformulated. Maybe the authors could use a terminology similar to Anderson (1976): “snow density change due to destructive metamorphism”

Introduction L 40-42: this one-sentence paragraph mentions the limitations in the representation of the hydrological cycle in land surface models. The authors simply mention that “some processes” are not well represented. I recommend them to list some of these processes in this paragraph and to add the corresponding references. In particular, they could provide more information about the processes related to cold regions hydrology. 

Introduction L 60: the authors could mention here that the multi-layer snowpack scheme used in EC-Land differs, in term of complexity, from detailed multi-layer snowpack schemes such as Crocus or Snowpack (see Boone and Etchevers, 2001 for a classification of the different types of snowpack schemes).  

Introduction L69: the authors could certainly rephrased this sentence since SWE, rather that snow depth, is the key variable in snow hydrology but I also understand the importance of accurately simulating snow depth to properly capture the impact of snow on the thermal regime of the underlying ground. At least, it would be good to explicitly mention SWE in this list.

P 3 L 102: the distinction between surface and subsurface runoff is important since it is discussed in the results section. It would be useful for the reader if the authors could add one or two sentences that explain how subsurface runoff is computed in EC Land (see also my comment below regarding surface runoff in presence of frozen soil).

P3 L 122: is there a representation of reservoir operation in the routing scheme (e.g. Yassin et al, 2019)? It would be interesting to provide the information here. 

P4 Eq (4): the left-term of this equation should be rewritten to clearly show that this equation gives the change in snow density as function of time. This equation is particularly important in this paper since the authors tested different values of the parameter $c$ used in the equation. In addition, the units of $b$ should be provided: K^{-1}.

P 5 L 185: as shown later in the results, the representation of the infiltration of liquid water into partially frozen soil is a key component of accurate hydrological simulations with EC Land. Therefore, the authors should add in this section a description of the parameterization of surface runoff in EC Land and how it is impacted by the presence of frozen water in the soil layers near the surface. The adjustments of runoff and percolation in presence of frozen water are generally crucial in land surface models and hydrological models to provide accurate streamflow simulations in cold regions (e.g. Niu and Yang, 2006; Koren et al., 2014). In particular, the soil freezing scheme, implemented in EC Land, has been initially developed to improve the simulation of stable boundary layers and differ from more advanced representation of soil freezing in land surface scheme that may be more suitable for hydrological simulations (e.g. Elshamy et al., 2020; Clark et al., 2021). This aspect of the model must be clearly explained in the paper and the potential impact of this simplified representation of soil freezing should be discussed.

P 5 L 206: it would be interesting for the reader to have a better idea of the distribution of catchment areas among the 453 stations that have been finally selected. Maybe the authors could add a figure that shows a histogram of the distribution of catchment areas.

P 6 L 226-227: the sensitivity analysis for the permafrost is carried out in a specific region that does not cover all the catchments that are potentially influenced by the presence of permafrost (as shown on Fig. 1). For example, stations located in Nunavut (Canada) seem to be excluded from the analysis. The authors should better justify their choice of region used in the sensitivity analysis.

P 9 L325-326: the value of $c$ will affect the densification rate of fresh snow but it will not influence the fact that the rate of change of snow density due to metamorphism is active only below 150 kg m-3. Indeed, this maximal value is imposed in Eq (4) based on the work of Anderson (1976). Therefore, I think that the sentence in the text should be rephrased.  

P 9 L L340: can the authors confirm that the formulation proposed in ML-Meta1 still includes the maximal value of 150 kg m-3?

P 11 L 409-411: in the Results section, the different subsections start with a sentence that summarizes the main conclusion of the subsection. Such approach is not common in scientific papers and usually the partial conclusions or key summary are given after the figures have been presented and discussed.  Maybe this approach is classic in Atmosphere and I just want to let the author know that I was surprised by this structure of the subsections. 

P 12: Fig. 4: How is computed snow melt on this figure? Is it the liquid water runoff at the base of the snow cover averaged over the catchment?

 P 12: Fig. 4: if possible, it would be very interesting to see the time series of the climatological mean of the frozen fraction of soil moisture in the top 100 cm of the soil. This would help to better understand how the changes in soil temperature in the experiment ML decrease the frozen fraction and increase the infiltration of liquid water from snowmelt in the soil column.

L 13 L 449-451: It would be interesting if the authors could add a sentence or two to describe the impact of land data assimilation on water conservation in the hydrological simulations. Is the low bias associated with removal of snow by the land data assimilation during the snow melt period? It is done at P 557-559 and I think it would be useful to add it here when talking about the impact of land data assimilation for the first time.    

P 13 L 455-457: this sentence is not clear and should certainly be rephrased. Indeed, I would expect that the fact that snow is melting later in the ML simulation (exposing bare ground to the atmosphere later) explains why the soil temperature starts its fast increase later in the ML simulation than in the SL simulation, and not the opposite.

P 15 L 497: The ML-Vert experiment uses a new formulation for the snow layering. It is active in regions of complex terrain and for snow depth larger than 25 cm. Therefore, it would be useful if the authors could mention which percentage of the catchment of the Olenesk river is impacted by the change in the vertical discretization.

P 16 L 564-565: based on Fig. 7, it seems that the improvements obtained with the ML-Opt configuration in the permafrost study area are not observed at the stations located in the Canadian Arctic (Nunavut mainly), outside of the permafrost study area. The hydrological cycle at these stations is still supposed to be impacted by the presence of permafrost. Can the author provide a potential explanation for this behavior? Is it associated with a physical process which is not well represented in EC Land and that may affect streamflow generation in this region?

P 17 L 581-585: Figure 7 illustrates well how the ML-Opt configuration affects the models performance when compared to the ML default configuration. For the other experiments, it only provides information about the aggregated statistics. If possible, I recommend to the authors to add similar maps that show the change in kge between some of the experiments and the reference ML configuration. These maps could be added as supplementary material. For example, it would allow the authors to discuss the impact of the new snow vertical discretization proposed in the ML-Vert configuration. Does this configuration improve hydrological simulations in mountainous catchments in the US and Canada?

P 17 L 583-584: can the authors provide more information about this positive bias around the Rockies? Indeed, I was more expecting a negative bias due to an underestimation of snow accumulation in the mountainous headwater catchments due to an underestimation of orographic enhancement in ERA5, typically found in global reanalysis (e.g., Wrzesien et al., 2019).

P 18 L 646-653: I recommend the authors to mention the processes that are not represented in EC land and that strongly influence hydrological simulation in cold regions (mass loss due to blowing snow sublimation, snow interception by high-vegetation and associated sublimation, proper representation of infiltration into frozen soil, …) (see for example Krogh et al., 2017).

Technical Comments

Text

P 3 L 107: can the authors give the model time step used in EC Land in this offline configuration?

P 9 L 337: add the unit m3 kg-1 for c.

Figures

P 10 Fig. 3: it would be useful for the reader to show on the maps c) and d) the variable –abias and –avar so that the meaning of the color scale would be the same among the four maps.

P 12 Fig. 4: the units for soil moisture should be revised.

References

Barrere, M., Domine, F., Decharme, B., Morin, S., Vionnet, V., & Lafaysse, M. (2017). Evaluating the performance of coupled snow–soil models in SURFEXv8 to simulate the permafrost thermal regime at a high Arctic site. Geoscientific Model Development, 10(9), 3461-3479.

Boone, A., & Etchevers, P. (2001). An intercomparison of three snow schemes of varying complexity coupled to the same land surface model: Local-scale evaluation at an Alpine site. Journal of Hydrometeorology, 2(4), 374-394.

Clark, M. P., Zolfaghari, R., Green, K. R., Trim, S., Knoben, W. J., Bennett, A., ... & Spiteri, R. J. (2021). The numerical implementation of land models: problem formulation and laugh tests. Journal of Hydrometeorology, 22(6), 1627-1648.

Domine, F., Picard, G., Morin, S., Barrere, M., Madore, J. B., & Langlois, A. (2019). Major issues in simulating some Arctic snowpack properties using current detailed snow physics models: Consequences for the thermal regime and water budget of permafrost. Journal of Advances in Modeling Earth Systems, 11(1), 34-44.

Elshamy, M. E., Princz, D., Sapriza-Azuri, G., Abdelhamed, M. S., Pietroniro, A., Wheater, H. S., & Razavi, S. (2020). On the configuration and initialization of a large-scale hydrological land surface model to represent permafrost. Hydrology and Earth System Sciences, 24(1), 349-379.

Gouttevin, I., Langer, M., Löwe, H., Boike, J., Proksch, M., & Schneebeli, M. (2018). Observation and modelling of snow at a polygonal tundra permafrost site: spatial variability and thermal implications. The Cryosphere, 12(11), 3693-3717.

Koren, V., Smith, M., & Cui, Z. (2014). Physically-based modifications to the Sacramento Soil Moisture Accounting model. Part A: Modeling the effects of frozen ground on the runoff generation process. Journal of hydrology, 519, 3475-3491.

Krogh, S. A., Pomeroy, J. W., & Marsh, P. (2017). Diagnosis of the hydrology of a small Arctic basin at the tundra-taiga transition using a physically based hydrological model. Journal of hydrology, 550, 685-703.

Niu, G. Y., & Yang, Z. L. (2006). Effects of frozen soil on snowmelt runoff and soil water storage at a continental scale. Journal of Hydrometeorology, 7(5), 937-952.

Niu, G. Y., & Yang, Z. L. (2007). An observation‐based formulation of snow cover fraction and its evaluation over large North American river basins. Journal of geophysical research: Atmospheres, 112(D21).

Royer, A., Picard, G., Vargel, C., Langlois, A., Gouttevin, I., & Dumont, M. (2021). Improved Simulation of Arctic Circumpolar Land Area Snow Properties and Soil Temperatures. Frontiers in Earth Science, 9, 515.

Wrzesien, M. L., Durand, M. T., and Pavelsky, T. M.: A reassessment of North American River basin cool-season precipitation: Developments from a new mountain climatology data set, Water Resour. Res., 55, 3502–3519, https://doi.org/10.1029/2018WR024106, 2019. 

Yassin, F., Razavi, S., Elshamy, M., Davison, B., Sapriza-Azuri, G., & Wheater, H. (2019). Representation and improved parameterization of reservoir operation in hydrological and land-surface models. Hydrology and Earth System Sciences, 23(9), 3735-3764.

Author Response

(The authors gave the same response as above.)

Round 2

Reviewer 2 Report

The authors have sufficiently addressed my previous comments. I recommend the article for publication.